# Performance Assessment and Comparison of Deployment Options for 5G Millimeter Wave Systems

**Evgeni Mokrov** [1,*,†] and **Konstantin Samouylov** [1,2,*,†]

1   Applied Mathematics and Communications Technology Institute, Peoples' Friendship University of Russia (RUDN University), 6 Miklukho-Maklaya St., 117198 Moscow, Russia
2   Institute of Informatics Problems, Federal Research Center Computer Science and Control of Russian Academy of Sciences, 119333 Moscow, Russia
*   Correspondence: mokrov-ev@rudn.ru (E.M.); samouylov-ke@rudn.ru (K.S.)
†   These authors contributed equally to this work.

**Abstract:** The roll-outs of fifth-generation (5G) New Radio (NR) systems operating in the millimeter-wave (mmWave) frequency band are essential for satisfying IMT-2020 requirements set forth by ITU-R in terms of the data rate at the access interface. To overcome mmWave-specific propagation phenomena, a number of radio access network densification options have been proposed, including a conventional base station (BS) as well as integrated access and backhaul (IAB) with terrestrial and aerial IAB nodes. The aim of this paper is to qualitatively and quantitatively compare the proposed deployments using coverage, spectral efficiency and BS density as the main metrics of interest. To this end, we develop a model capturing the specifics of various deployment options. Our numerical results demonstrate that, while the implementation of terrestrial relaying nodes potentially improves coverage and spectral efficiency, aerial relays provide the highest coverage, three times that of a direct link connection, and also significantly reduce the required BS density. The main benefit is provided by the link between the BS and the aerial relay. However, gains are highly dependent on a number of elements in antenna arrays and targeted outage probability. The use of terrestrial relays can be considered a natural trade-off between coverage and the aggregate rate.

**Keywords:** 5G; New Radio; IAB; UAV; performance comparison

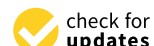



## 1. Introduction

As fifth-generation (5G) New Radio (NR) systems operating in both sub-6 GHz and millimeter-wave (mmWave) bands have become standardized, operators have started to deploy them [1,2]. However, the roll-outs of mmWave NR are hampered by the propagation specifics of this band [3]. These include much higher path losses compared to microwave ($\mu$Wave) systems [4,5], blockage of propagation paths between the base station (BS) and user equipment (UE) by small dynamic [6,7] and large static objects [8], as well as the micromobility of UEs [9,10]. These effects drastically reduce the coverage of a single 5G mmWave NR BS. As a result, deployment of 5G mmWave NR technology that may finally satisfy IMT-2020 requirements in terms of the data rate at the access interface requires extremely dense deployments of 5G mmWave NR BSs, resulting in overwhelming capital expenditures (CAPEX) for network operators.

As a way to provide cost-efficient densification of 5G mmWave NR systems, 3GPP has recently proposed integrated access and backhaul architecture (IAB). By utilizing multi-hop communications via so-called IAB nodes characterized by reduced equipment complexity compared to conventional BSs, called IAB donors, these systems bring the network access points closer to the users. They allow for efficient alleviation of the frequency of blockage situations that has detrimental effects on user session continuity [11–13].

The initial specifications in Release 16 [14] advocate IAB architecture for stationary terrestrial IAB nodes. However, this solution is rather limited in dense irregular city

deployments as there should always be line-of-sight (LoS) conditions between IAB donors and IAB nodes or between two IAB nodes [15]. Subsequently, in Release 17 [16], where the architecture has been further refined, the use of mobile IAB nodes that might be installed in cars or unmanned aerial vehicles (UAV) is proposed. The use of UAVs has recently been shown to be very useful as network access points at high altitude, enabling efficient avoidance of blockage by large static objects, such as buildings.

In spite of different types of deployment options for 5G mmWave NR systems, including conventional BS-based and terrestrial [17] or mobile IAB systems, to the best of the authors' knowledge, there has been no detailed assessment as to which provides the best performance for these systems under different environmental conditions. On the one hand, the use of UAV-mounted IAB nodes brings high blockage tolerance at the expense of additional operational expenditures (OPEX) for the network operators and longer propagation paths [18]. On the other hand, in some deployment conditions, such as suburbs and rural areas, terrestrial, and even conventional, BS-based roll-outs might be sufficient. In our paper, we qualitatively and quantitatively characterize different types of 5G mmWave NR deployment options, including (i) conventional BS-based systems, (ii) terrestrial IAB systems, and (iii) aerial IAB systems. Similarly, terrestrial coverage was studied in [19]; however, the study focused on transmission power impact to coverage, whereas we consider antenna arrays and outage probability as parameters and also give a comparison against aerial IABs. We account for specific mmWave propagation conditions, including blockage by small dynamic and large static objects. The main metrics of interest are coverage and spectral efficiency.

The rest of the paper is organized as follows: In Section 2, we introduce our system model, describe the considered scenarios of direct connection and the terrestrial and aerial relays, and introduce blockage models according to the 3GPP standard [8]. In this section, we also describe the main metrics of interest. Here, similarly to [20], we focus on an analysis of spectral efficiency and coverage. Then, in Section 3, we establish coverage and spectral efficiency metrics for the considered deployment options. The numerical results are presented in Section 4. We show that the main benefit is achieved for the link from BS to UAV, giving three times the coverage of direct connection. In addition, we highlight the drawbacks of using aerial relays, since it is necessary to increase the number of antenna elements on the relay in order to reduce outage probability. Finally, we calculate the BS density metric to evaluate the quantitative advantage aerial relays provide compared to other scenarios in terms of the BS cells necessary to cover a given area. Finally, conclusions are drawn in the last section.

## 2. System Model

In this section, we introduce our system model. We subsequently define deployment, propagation, and blockage models. Finally, we define our metrics of interest.

### 2.1. Deployment

We consider a scenario where a single BS with circular coverage provides service to the users in a crowded environment in urban conditions (Figure 1). The radius of the coverage area of the cell BS $r_{BS}$ depends on the antennas used, both on the UE and BS, as well as their height difference and targeted outage probability $P_0$, i.e., the probability of signal loss that is deemed acceptable and is computed using the propagation model presented below. Since we consider an urban scenario, the signal that the BS sends to the UE can be blocked by both large static buildings and other small mobile user bodies. The heights of the BS and UE are assumed to be constant and given by $h_{BS}$ and $h_{UE}$, respectively. In order to enhance the coverage area, we consider and compare two cases of using relays, located at height $h_{Ter}$ in the case of a terrestrial relay, or $h_{UAV}$ in the case of an aerial mobile relay located on the UAV (Figure 1). In both these cases, the signal from the BS is first sent to the relay node and then forwarded by the relay to the UE. Thus, the coverage areas are calculated for both the BS and relay nodes. The resulting coverage consists of the sum of both coverages from

BS to the relay and from the relay to the UE. We consider that the BS uses a single output antenna, while, in the case of direct connection, the UE uses a MIMO antenna array with *k* elements. In scenarios with relays, the UE has a fixed $4 \times 4$ MIMO antenna, while the relays have MIMO antenna arrays with *k* elements. This assumption is made in order to investigate the impact the number of elements has on the considered metrics by varying the parameter *k*. Since we mainly focus on the signal received by the UE, we can safely ignore the number of transmitting antenna elements on the UE and only vary the number of transmitting and receiving antenna elements on the relay and the number of receiving antenna elements on the UE. We have to consider a number of transmitting and receiving antenna elements on the relay since it receives the signal from the BS and forwards it to the UE. However, to keep the plots two-dimensional, we only vary the number of receiving antenna elements on the UE in the case of direct connection with the BS, or the number of receiving antenna elements on the relay, while fixing all other numbers of antenna elements as constants.

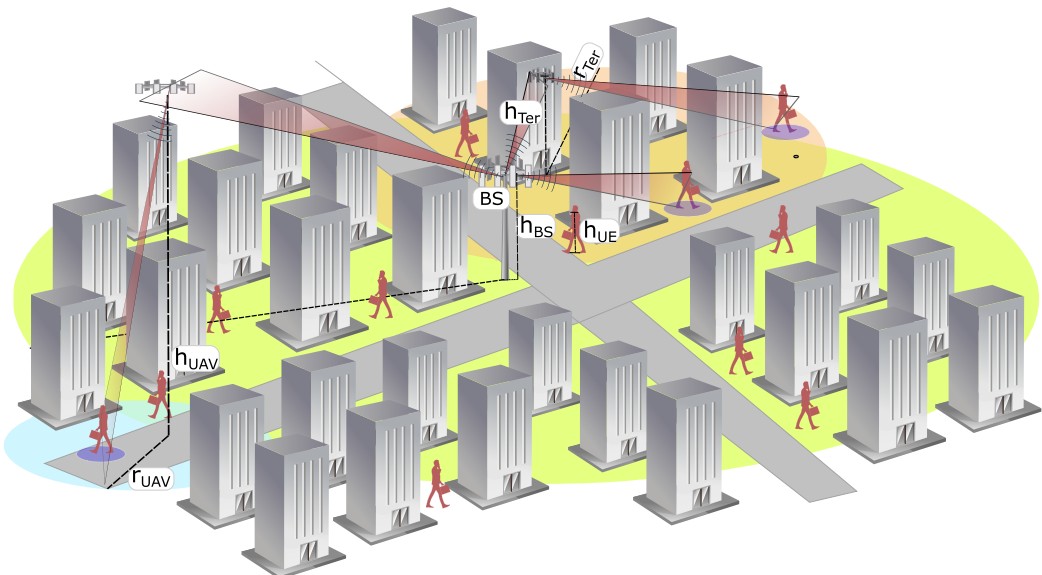

**Figure 1.** The considered deployment scenarios.

## 2.2. Blockage Models

We consider the scenario where the signal from the transmitter located at height $h_{tx}$ to the receiver, located at height $h_{rx}$, can be blocked by both large static blockers, such as buildings, and small dynamic blockers, such as people. We consider different models for both blockers and then calculate the total blockage probability.

There are a number of blockage models proposed to date including those related to human body blockage [21] and LoS/NLoS link behavior [22,23]. To calculate the human body blockage probability by user blockers, we use the human blockage model from [21], where human blockers are modeled as cylinders with radius and height $r_B$ and $h_B$, respectively. They are placed with density $\lambda$ units/m$^2$ and the blockage probability is given by (1)

$$p(l) = 1 - exp\left(-2\lambda r_B \left[\sqrt{l^2 - (h_{tx} - h_{rx})^2} \frac{h_B - h_{rx}}{h_{tx} - h_{rx}} + r_B\right]\right). \tag{1}$$

Here, $h_{tx}$ is the height of the transmitter, $h_{rx}$ is the height of the receiver, and *l* is the three-dimensional distance between the transmitter and the receiver.

To calculate the LOS probability for large stationary buildings, we implement 3GPP path-loss propagation models for UMi (2), UMa (3), with parameters given in (4), and RMa (5) scenarios ([8])

$$p_{LOS}(l) = \begin{cases} \left[\frac{18}{l} + exp(-\frac{l}{36})(1 - \frac{18}{l})\right] & 18\ m < l \\ 1 & otherwise \end{cases}, \tag{2}$$

$$p_{LOS}(l) = \begin{cases} \left[\frac{18}{l} + exp(-\frac{l}{63})(1 - \frac{18}{l})\right]\left(1 + C'(h_{rx})\frac{5}{4}(\frac{l}{100})^3 exp(-\frac{l}{150})\right) & 18\ m < l \\ 1 & otherwise \end{cases}, \tag{3}$$

$$C'(h) = \begin{cases} \left(\frac{h-13}{10}\right)^{1.5} & 13\ m < h \le 23\ m \\ 0 & h \le 13\ m \end{cases}, \tag{4}$$

$$p_{LOS}(l) = \begin{cases} exp\left(-\frac{l-10}{1000}\right) & 18\ m < l \\ 1 & otherwise \end{cases}. \tag{5}$$

According to the 3GPP Urban Micro model (UMi) [8], the path loss of a signal transmitted at a carrier frequency $f$, at a given three-dimensional distance $l$, can be calculated as

$$PL(l, \alpha, \beta) = 10\alpha log_{10}(l) + 10\beta + 20log_{10}(f), \tag{6}$$

where the coefficients $\alpha$ and $\beta$ are responsible for human and building blockage, respectively, are given in Table 1. In addition, we consider the signal to be blocked by both small mobile blockers and large stationary blockers with a certain probability in all cases, except the case of a BS-to-UAV connection. In that case, since the UAV is located high above the ground and the BS antenna is also located higher than a human height, for this link, we assume that human blockage is not possible and only consider blockage by large stationary buildings; thus, in this case, we only use value $\beta_1 = 3.24$ from Table 1.

**Table 1.** LoS coefficients.

| Coefficient | LoS Value | NLoS Value | Description |
|---|---|---|---|
| $\alpha$ | 2.1 | 3.19 | Building blockage coefficient |
| $\beta$ | 3.24 | 5.24 | Human blockage coefficient |

### 2.3. Metrics of Interest

In the considered system, we are interested in the cell coverage and efficiency in the case of a direct connection versus use of terrestrial and aerial (UAV) re-transmitters in an urban environment for a sub-mmWave network. Specifically, we are interested in such metrics as the spectral and the spatial efficiency (16) and (17), to estimate the channel transmission rate and coverage, and the BS density (18), to compare the number of BSs necessary to serve a $r_0 = 1$ km area for direct and indirect connection.

## 3. Analytical Model

For the considered scenario, we calculate the signal-to-noise ratio (SNR) at distance $l$ m from the transmitter as (7)

$$SNR(l, P_0) = 10Log_{10}P + G_{tx} + G_{rx} - N - SNR_{marg} - SFM(P_0) + SCM$$

$$- 10lg\left(\sum_{i\in\{0,1\}}\sum_{j\in\{0,1\}}|1 - i - p_{LoS}(l)||1 - j - p_l(l)|PL(l, \alpha_i, \beta_j)\right), \tag{7}$$

where $P$ is the signal power in Wt, and $G_{tx}$ and $G_{rx}$ are gains in dB of the transmitter and the receiver, respectively, and $N$ is the noise level in dB. Here $\alpha_i$, $\beta_j$ are the blockage coefficients from Table 1, and, thus, do not have dimension, $i = 0$ stands for the NLoS case, $i = 1$ stands for the LoS case for building blockages, and the same holds true for $j$ in the case of human blockages. $SNR_{marg}$ is the MCS margin in dB, given in accordance with [24]. This margin is implemented to take into account a situation where the signal is low enough to reach MCS 16, resulting in signal loss. $SCM$ is Shannon's capacity margin in dB, implemented to mitigate inaccuracy occurring due to using theoretical values derived from Shannon's theorem, $P_0$ is the outage probability, i.e., the acceptable probability of connection loss, and $SFM$, calculated as (8), is the slow fading margin in dB. This margin takes into account possible signal variations due to the slow fading effect since we consider scenarios with building blockages. The latter parameter is given by

$$SFM(P_0) = \sigma_{SF}\sqrt{2}Errf^{-1}(2P_0), \tag{8}$$

$SNR_{marg}$ is the SNR value margin, and $\sigma_{SF}$ is the shadow fading parameter dependent on the scenario.

Here, we also consider a composite noise model, where the noise level is calculated as the sum of the internal and external noise, according to (9), i.e.,

$$N = N_{int} + N_{ext}. \tag{9}$$

Internal noise can be caused by cable losses, i.e., signal loss due to the cable length $loss_c$, and the noise figure $NF_{UE}$, defined by the UE device internal configuration. Thus, internal noise can be calculated as the sum of its components, as shown in (10), i.e.,

$$N_{int} = loss_c + NF_{UE}. \tag{10}$$

External noise sources (11) include thermal noise $N_0$ and the interference margin $I_{Marg}$ introduced to consider the worst-case scenario of signal interference between two different transmitters occurring on the cell edge.

$$N_{ext} = N_0 + I_{marg}. \tag{11}$$

After implementing the blockage model, as specified in Section 2, we can calculate the transmitter coverage area with gain $G_{tx}$ located at height $h_{tx}$ that transmits a signal at a carrier frequency $f$ to a receiver located at height $h_{rx}$ with gain $G_{rx}$. The coverage would be a three-dimensional distance from the transmitter to the receiver and, thus, is given by (12)

$$r = \sqrt{l^2 - (h_{tx} - h_{rx})^2}, \tag{12}$$

where $l$ is the distance between the transmitter and the receiver. Using (6), we can obtain this distance $l$ as (13)

$$l = \left(\frac{10^{PL}}{10^{\beta}f^2}\right)^{\frac{1}{\alpha}}. \tag{13}$$

Then, we can substitute the path loss from (7) for the worst case of NLoS and, thus, we get (14)

$$l = \left(\frac{P10^{G_{tx}+G_{rx}-N-SNR_{marg}-SF+SCM}}{10^{\beta}f^2}\right)^{\frac{1}{\alpha}}. \tag{14}$$

From (12) and (14), and fully expanding $N$, we can express the coverage area as (15)

$$r = \sqrt{\left(\left(\frac{P10^{\frac{G_{tx}+G_Rx}{10}}}{10^{\beta}f^2 10^{\frac{loss_c+NF_{UE}+N_0+I_{Marg}+SNR_{marg}-SCM+SFMnlos}{10}}}\right)^{\frac{2}{\alpha}} - (h_{tx}-h_{rx})^2\right)} \qquad (15)$$

Recall that the spectral efficiency is given by (16), while the spatial efficiency is given by (17)

$$E(l) = log_2(1 + SNR(l)), \qquad (16)$$

$$E_S(l,r) = 10^6 \frac{E(l)}{\pi r^2}. \qquad (17)$$

Here, in order to calculate the spatial efficiency, we calculate the coverage area as a circle around the base station with an effective transmission radius $r$ m. A multiplier $10^6$ is used to convert the radius $r$ into km.

The last metric we focus on is the BS density, which is calculated as the number of cells necessary to cover the same surface area as a circle with radius $r_0$ (18). This metric shows the number of BSs needed to cover an area and can be used to plan network topology.

$$n(l) = \frac{2\pi(r_0-1)^2}{3\sqrt{3}l^2}; \qquad (18)$$

## 4. Numerical Results

In this section, we will elaborate our numerical results by assessing the performance of different 5G mmWave deployment options: (i) conventional BS-only based, (ii) terrestrial IAB deployment, and (iii) airborne IAB deployment utilizing UAV as IAB nodes. As metrics of interest, we utilize the coverage of a single BS, spectral efficiency, and the required BS deployment density.

The default system parameters are provided in Table 2. We consider links from the aerial IAB to the UE to use the UMa scenario, while other links use the UMi scenario. This consideration is due to the fact that the aerial IAB is located at a high altitude and, thus, is located above rooftop level, while the BS, terrestrial IAB, and UEs are all located close to ground level in urban areas. The scenario parameters are given in Table 3.

**Table 2.** System parameters.

| Parameter | Value | Description |
|---|---|---|
| $\alpha_0$ | 3.19 | NLoS building blockage coefficient |
| $\alpha_1$ | 2.1 | LoS building blockage coefficient |
| $\beta_0$ | 5.24 | NLoS human blockage coefficient |
| $\beta_1$ | 3.24 | LoS human blockage coefficient |
| $f$ | 28 Ghz | Carrier frequency |
| $P$ | 2 Wt | Transmitted signal power |
| $SNR_{marg}$ | $-9.478$ dB | Signal-to-noise ratio margin |
| $SCM$ | 3 dB | Shannon capacity margin |
| $P_0$ | 0.1, 0.01 | Outage probability |
| $loss_C$ | 2 dB | Cable loss |
| $NF_{UE}$ | 7 dB | Noise figure due to UE configuration |
| $r_0$ | 1 km | Base radius |
| $h_{BS}$ | 10 m | eNB height |
| $h_{Terr}$ | 10 m | Terrastial re-transmitter height |
| $h_{UE}$ | 1.3 m | User equipment height |
| $h_{UAV}$ | 100 m | UAV re-transmitter height |

**Table 2.** *Cont.*

| Parameter | Value | Description |
|-----------|-------|-------------|
| $\lambda$ | 0.3 units/m$^2$ | Density of human blockers |
| $h_B$ | 1.7 m | Human blocker height |
| $r_B$ | 0.3 m; | Human blocker radius |
| $G_{BS}$ | 14.58 dB | eNB antenna gain |
| $G_{Terr}$ | 14.58 dB | Terrastial re-transmitter gain |
| $G_{UE}$ | 5.57 dB | User equipment antenna gain |

**Table 3.** Deployment scenarios.

| Scenario | Path Loss | Shadow Fading | Applicable Heights |
|----------|-----------|---------------|--------------------|
| UMi | (2) | $\sigma_{SF} = 4, \sigma_{NLOS} = 7.82$ | $h_{BS} = 10$ m |
| UMa | (3)–(4) | $\sigma_{SF} = 4, \sigma_{NLOS} = 6$ | $h_{BS} = 10$ m |

The rest of this section is organized as follows: We start with an assessment of the outage probability and spectral efficiency for urban deployment conditions. Then, we compare the coverage areas and BS placement density necessary to cover a given area for all options.

### 4.1. Spectral Efficiency

We start our analysis in Figure 2 illustrating the LoS probability (Figure 2a) and the spectral efficiency (Figure 2b) as a function of the distance between the UE and BS for three considered deployment options: (i) BS-UE, (ii) BS-UAV-UE, and (iii) BS-Terr-UE, where *Terr* implies terrestrial IAB node deployment. For this and further analyses, we consider different antenna arrays, but, since the plot behaviors for different arrays were similar, we present those that give better insight into the impact the number of elements has on the discussed metrics.

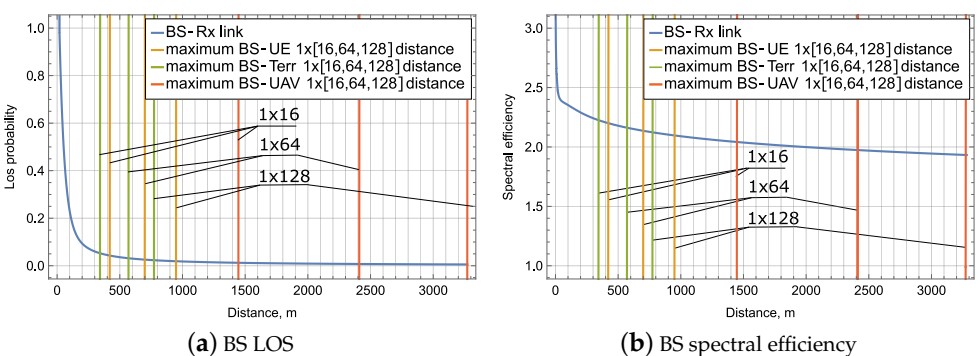

(**a**) BS LOS          (**b**) BS spectral efficiency

**Figure 2.** LOS and spectral efficiency as a function of distance.

Both the LoS probability and spectral efficiency drop with distance. The vertical lines represent the maximum distance at which corresponding devices (i.e., UE, terrestrial and aerial re-transmitters), equipped with antenna arrays mentioned in the plot legend element numbers, can be placed, while maintaining an outage probability of 0.1. It is evident that, while the terrestrial re-transmitter can be placed further than the UE by several tens of meters, the aerial UAV gives a much higher advantage in terms of a distance of 1–2 km.

### 4.2. Coverage Metrics

We now proceed to understand the coverage of the considered deployment options illustrated in Figure 3 for different deployment options and outage probabilities. Here, Figure 3b shows the coverage of individual links, while Figure 3a presents the full coverage

of the considered options. Note that the target end-to-end outage probability is set at $P_0 = 0.1$ and, thus, for the deployment options in Figure 3b involving more than one link, the target outage is set to 0.05. In Figure 3b the best coverage is produced by the BS-UAV link, depicted as a blue line. The second best link is the violet direct link BS-UE, closely followed by the green link from the BS to the terrestrial re-transmitter; the links from the terrestrial (bright red) and aerial (light orange) re-transmitters to UE show the lowest coverage. These result in full coverage having a similar order: the best coverage is provided using an aerial re-transmitter with an outage probability of 0.1 (blue line), followed by the same scenario with an outage of 0.01 (red line). However, even with stricter conditions on the outage probability, the aerial IaB still shows better results than a terrestrial aerial for both 0.1 (yellow line) and 0.01 (brown line) outage probabilities. The lowest coverage for both probabilities has a direct connection; however, while under $P_0 = 0.01$, it is the lowest (brown) plot line, in the case of $P_0 = 0.1$ (green line), it gives better results than the terrestrial IaB under a stricter outage probability of 0.01. By analyzing the data presented in the former illustration, we observe that the best coverage is provided by the BS-UAV link, with the gains over the second most distant link, the BS-UE link, being of the order of 300%. The rationale is that, in the considered urban deployment conditions, the average height of the building is much greater than the height of the LoS link and, thus, it is limited by LoS path loss only. Note that both links are unaffected by the dynamic human body blockage and, thus, the difference is solely due to building blockage impairments.

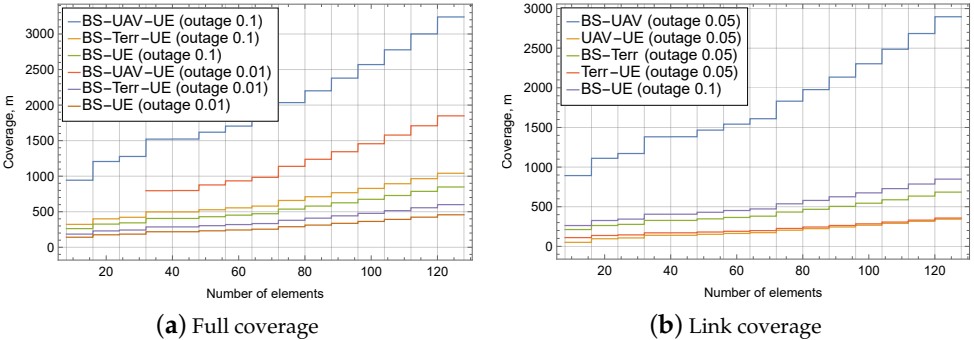

(**a**) Full coverage                                                                    (**b**) Link coverage

**Figure 3.** Single BS coverage as a function of deployment option.

Figure 4 also presents full coverage (Figure 4a) and individual links coverage (Figure 4b) for all three considered options. In Figure 4a, blue lines indicate a link through an aerial relay, yellow lines, a link through a terrestrial relay, while green lines correspond to a direct connection. In Figure 4b, blue lines indicate a link between BS and the aerial relay, yellow lines, a link from the aerial relay to the UE, green lines, a link between the BS and the terrestrial relay, red lines, a link from the terrestrial relay to the UE, and the violet lines correspond to a direct connection. The numbers on the plots indicate the number of elements used in the antenna arrays on relays or the case of direct connection on the UE. Here, we study the impact of the outage probability on the coverage. It is evident from the figure that, the more elements are used in the antenna array, the higher the impact of the outage probability is. For less than 16 element links from the BS to the UE, directly or via a terrestrial re-transmitter, very small changes of about 200 m are experienced, while, in the case of the aerial re-transmitter, it reaches 500 m. The plot in Figure 4a for $P_0 \leq 0.02$ stays at 0 because it is impossible to reach the UE from the aerial re-transmitter location with the given outage probability and a number of antenna elements. This is also reflected in Figure 4b, where the link between UAV-UE starts from this point. Since the resulting coverage is a sum of the link coverages, we have a sharp rise in Figure 4a. Similar behavior can be seen in Figure 4a, where, for the UAV-UE link and $P_0 = 0.01$, the plot starts only from 36 antenna elements. This is also due to the fact that, with fewer elements, it is impossible to reach the UE with a given outage probability. The fact that the plot does not slowly rise

from 0, as it does in Figure 4b, is bcause the number of elements is a discrete parameter, resulting in the plot involving a stair-step.

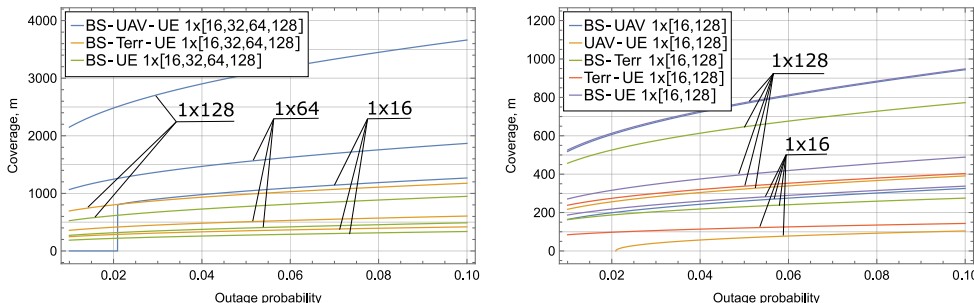

(**a**) Full coverage as a function of outage probability  (**b**) Link coverage as a function of outage probability

**Figure 4.** Coverage as a function of outage probability.

Interestingly, the UAV-UE and Terr-UE nodes are characterized by approximately the same coverage despite the difference between the terrestrial and UAV IAB node deployments. A plausible explanation is that the height of the terrestrial IAB node (10 m) is already high enough to mitigate dynamic human body blockage situations, while impairments induced by the IAB node height are very similar due to the low height of the UE. Thus, we can conclude that the height of the lowest communications entity is the one defining the coverage radius of a link.

The final intermediate metric affecting the BS deployment density is the end-to-end spectral efficiency and the spatial spectral efficiency illustrated in Figure 5b and Figure 5a, respectively, for an end-to-end outage probability $P_0 = 0.1$. By analyzing the spectral efficiencies, we observe that the difference between deployment options is almost unnoticeable. The rationale is that all of them include a link to the UE that is heavily affected by both dynamic human body blockage and building blockage and determines the end-to-end spectral efficiency. However, as can be seen in Figure 5b, the difference in spatial efficiency is drastic, with the BS-UAV-UE option characterized by the least performance. Interestingly, the use of terrestrial BS in the BS-Terr-UE deployment option is characterized by the highest possible performance despite having a large coverage, as demonstrated in Figure 3. This terrestrial deployment is a natural trade-off between the coverage and the aggregate rate.

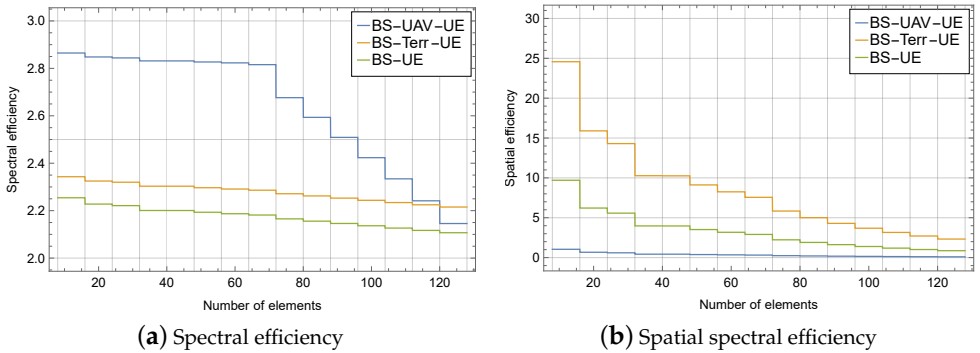

(**a**) Spectral efficiency  (**b**) Spatial spectral efficiency

**Figure 5.** Spectral and spatial efficiencies of the considered deployment options.

Finally, concluding the numerical results, we assess the required density of the BS in Figure 6. Here, Figure 6a shows the BS density for two end-to-end outage probabilities 0.1 and 0.01, while Figure 6b illustrates it for antenna arrays utilized at the link to the BS. Analyzing the data, one may observe that the required BS density depends heavily on the antenna array and outage. For a smaller number of elements, e.g., 20–40, the difference in the BS density between end-to-end outage 0.1 and 0.01 reaches 2–3 times, while, for a

higher number of elements, e.g., 100–120, all the deployments are principally similar in terms of BS density.

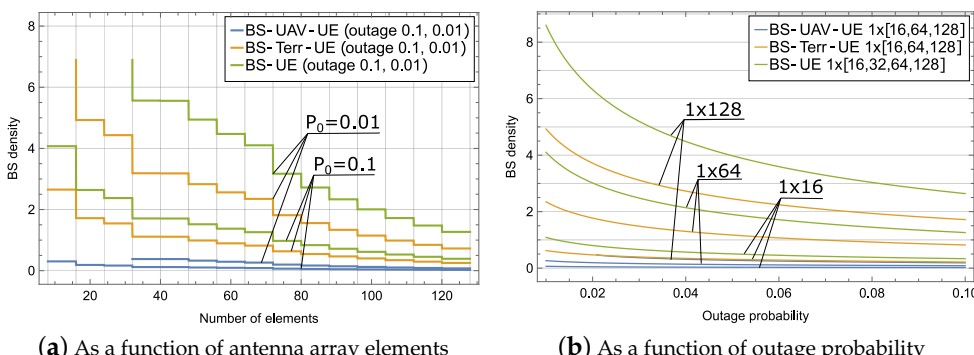

(**a**) As a function of antenna array elements　　(**b**) As a function of outage probability

**Figure 6.** BS density.

From the analysis provided, one can see that, although implementing UAVs as relays gives great benefits, nevertheless, these benefits have their own limitations. As was shown in this section, UAVs have a much higher outage probability than terrestrial relays with the same number of antenna elements. This limitation can be mitigated by using antennas with more elements; however, it would also impact the cost of such a system. However, with similar antennas, all the other parameters are much higher than those of terrestrial relays and direct connection. Thus, UAVs might provide the most benefits in large areas with relatively lax outage constraints, though, in small areas with a critical traffic flow, direct connection might be preferred, since there is not much difference in the studied metrics between a direct connection and terrestrial relays with a low number of antenna elements. Terrestrial relays with a large number of antenna elements might prove beneficial in scenarios where the area is not large enough for a UAV to achieve sufficient benefits to justify a higher number of antenna elements, but is sufficiently large for there to be enough of a difference between a direct connection and a terrestrial relay in terms of coverage.

## 5. Conclusions

In this work, we introduced system, propagation, and blockage models for different deployments for the 3gpp standard [8]. We studied three different scenarios of connection between a BS and a UE, including direct connection, connection via a terrestrial re-transmitter, and connection via an aerial re-transmitter. Our numerical results demonstrate that the aerial re-transmitters enable a single BS to serve an area of 1 km$^2$. In contrast, terrestrial re-transmitters reach this efficiency only with a significantly high number of elements in the antenna array— 64 elements for $P_0 = 0.1$, or even higher for a lower outage probability. At the same time, we showed that the most benefits are drawn from the link BS-UAV. We also identified a drawback in using aerial relays—the fact that the link UAV-UE is vulnerable to a low outage probability, and, in order to successfully operate under strict conditions on blockage probability, the number of antenna elements on the relay should be increased as, otherwise, the relay cannot hold a connection to the UE with a given outage probability. The vulnerable part is the link UAV-UE. At the same time, it was shown that terrestrial re-transmitters do not have that drawback, since they are located much closer to the UEs; however, it is much harder to provide the same coverage using them. One possible solution is the use of a hybrid system, where terrestrial re-transmitters would help to improve the vulnerable links in aerial relays; however, further study is necessary to evaluate the benefits and drawbacks of such a system.

Since, in the current work, we only considered data-link level with full-buffer traffic, in future work, it might be necessary to also compare these scenarios with regards to the number of users and their traffic types in relation to different architectures. Moreover, the

scenarios might be further improved by adding some different blocker types, such as cars and greenery.

**Author Contributions:** Conceptualization, K.S.; methodology, E.M.; software, validation and formal analysis, E.M.; writing—original draft preparation, E.M.; writing—review and editing, K.S.; supervision, K.S.; project administration, K.S.; funding acquisition, E.M. All authors have read and agreed to the published version of the manuscript.

**Funding:** The research was funded by the Russian Science Foundation (project No. 21-79-00157).

**Institutional Review Board Statement:** Not applicable.

**Informed Consent Statement:** Not applicable.

**Data Availability Statement:** Not applicable.

**Conflicts of Interest:** The authors declare no conflict of interest.

## Abbreviations

The following abbreviations are used in this manuscript:

| | |
|---|---|
| 5G | Fifth-Generation |
| NR | New Radio |
| CAPEX | Capital Expenditures |
| OPEX | Operational Expenditures |
| IMT | International Mobile Telecommunications |
| MIMO | Multiple-Input and Multiple-Output |
| IAB | Integrated Access and Backhaul |
| BS | Base Station |
| UAV | Unmanned Aerial Vehicle |
| UE | User Equipment |
| Terr | Terrestrial re-translator |
| SNR | Signal-to-Noise Ratio |
| UMa | Urban Macro Cell |
| UMi | Urban Micro Cell |
| RMa | Rural Macro Cell |
| Tx | Transmitter |
| Rx | Receiver |
| LoS | Line-of-Sight |
| NLoS | Non-Line-of-Sight |
| SCM | Shannon's Capacity Margin |
| MCS | Modulation and Coding Scheme |

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
