# Peer review of "Performance Assessment and Comparison of Deployment Options for 5G Millimeter Wave Systems"

_futureinternet, doi:10.3390/fi15020060_

Round 1

Reviewer 1 Report

Title: Performance Assessment and Comparison of Deployments Options for 5G Millimeter Wave Systems

The paper studies different network topology options in 5G mm-wave scenarios while considering urban micro, macro and rural scenarios. The results are presented in terms of spectral and spatial efficiency. While the topic of research is relevant and interesting, the way the paper is presented is not up to the mark, although the results seem interesting. The paper is poorly articulated and hence needs a major revision before it can be reconsidered. Some comments are as follows:

·         The discussion regarding equation (7) is not clear. What parameters are in dB and what parameters are taken in linear scale? Authors should clarify

·         Many acronyms have been used without being first defined such as MCS

·         The outage probability P0 has been discussed in context of eq. 7, but is not present in the equation

·         The reason for considering MCS margin and SFM margin are not clear, the context is missing

·         How is the noise power and interfering power modelled? For example, are they considered to be random like AWGN noise?

·         References for eq. 16 and 17 have not been provided.

·         Intent and context of using eq. 18 is not clear.

·         Fig. 2 discussion seems incomplete with a lot of things present in the figure have not been discussed, for example, why do you consider 1x16, 1x64 etc. scenarios.  It is not explicitly clear

·         Fig. 3 is not lucid enough, authors should choose better color line style and thickness to demarcate different cases. It is hard to comprehend.

·         Again, due to the formatting issues with Figures, the discussion on Figure 4,5 is really hard to follow. These figures should be formed properly for comprehension. Fig. 6 has better clarity.

·         Line 143, do authors mean P0 <=  0.02?

Author Response

 Thank you for your high evaluation of our work. We are sure your comments greatly benefitted our work. We tried our best to eliminate all the indicated flaws, that were indicated in the review. The focused point-to-point answers to the review comments follow in the attachment, where the feedback from the Reviewer is given in regular font, while our comments are highlighted in bold font.

Reviewer 2 Report

Authors compared the proposed deployments using coverage and spectral efficiency in 5G Millimeter wave systems. However before acceptance, major corrections are required:

1.     In abstract, please clearly specify the Purpose, Contribution, and findings.

2.     The overall flow is not clear highlight the major contribution in the contribution part of Introduction section.

3.     A small portion added in Ref. [16], please compare the proposed model and Ref. [16] in results section.

4.     How to find eq. (1), please provide the reference no or full derivation.

5.     Please improvement 2.1. Deployment. Is it MIMO or not clear. How many antenna use UE or BS.

6.     Please compare the proposed system results with previous one.

7.     Conclusion is not good. Gives some valuable comments about the paper.

8.     It is recommended to use a professional proofread and native English correction.

Author Response

(The authors gave the same response as above.)

Reviewer 3 Report

Please - see the attachment.

Author Response

Thank you for your interest in our work. Thanks to your comments we managed to add some additional references to improve our work.

Regarding the indoor scenario, unfortunately the focus of this work was on outdoor deployment, however investigating indoor deployment can be a good idea for further studies. When we commit to it we would also be able to consider different materials according to floor plans as well as small obstructions, such as potted plants. that way it would be possible to verify the derived data by measuring signal strength in the indoor case. In our opinion, such research would be definitely relevant, however, dur to its complexity and scale should be a separate work, continuing the current one.

Round 2

Reviewer 1 Report

All the comments have been addressed. Thank you for making all the changes.

Author Response

Thank you for you help in improving our work!

Reviewer 2 Report

Thanks for corrections. 

Author Response

(The authors gave the same response as above.)

Reviewer 3 Report

The paper discusses the deployment of 5G New Radio (NR) systems in the millimeter wave (mmWave) frequency band to meet the data rate requirements set forth by ITU-R. To overcome mmWave-specific propagation phenomena, several radio access network densification options have been proposed, including conventional base station (BS) based and integrated access and backhaul (IAB) with terrestrial and aerial IAB nodes.

The paper aims to compare these proposed deployments in terms of coverage, spectral efficiency, and BS density. To do this, the authors develop a model capturing the specifics of various deployment options and present numerical results. The results show that while terrestrial relaying nodes improve coverage and spectral efficiency, aerial relays provide the highest coverage and reduce required BS density.

One way to improve the paper would be to provide more detail on the specific implementation of the proposed deployment options. Additionally, the paper could provide more concrete examples of how the different densification options would be applied in practice. It would be more informative if the paper would also provide more detail on the different densification options, including their practical implementation and the necessary infrastructure, cost and the trade-off with benefits. Also, the paper would benefit from more discussion and analysis of the results and the limitation of the study, rather than just listing the results. It is also recommended to add more realistic and practical simulation results and performance metrics as well as real-world experimental results, to provide more concrete evidence of the proposed method's effectiveness. It would be helpful if the paper also discusses future work and the potential impact of the proposed deployment options on the 5G network infrastructure and operation. Finally, citing related studies and references in the literature on mmWave radio access network densification and 5G systems would provide a broader context for the proposed deployments and the results presented in the paper.

- “5G NR: The Next Generation Wireless Access Technology” by E. Beneat, J. de Rosny, et al. “Millimeter-Wave Mobile Communications for 5G Cellular: It Will Work!” by R. W. Heath Jr., T. S. Rappaport, et al.

- I/Q Imbalance Aware Nonlinear Wireless-Powered Relaying of B5G Networks: Security and Reliability Analysis," IEEE Transactions on Network Science and Engineering 8 (4), 2995-3008 , 2021.

- MGR: Multi-parameter Green Reliable communication for Internet of Things in 5G network", Journal of Parallel and Distributed Computing 118, 34-45, 2018.

- “3GPP TR 38.901 version 15.1.0, Study on Channel Model for Frequencies from 0.5 to 100 GHz” for the latest information about 3GPP channel models for 5G.

- 5G-enabled Hierarchical architecture for software-defined intelligent transportation system", Computer Networks 150, 81-89, 2019

Author Response

Thank you for your comments, we tried to edit article in a way to resolve the issues you mentioned:

  • We added some additional details to better describe the deployment options
  • We added more discussion about the results in the System model section
  • Regarding practical trial, unfortunately UAV and IaB technologies are still not implemented by the mobile operators, making it impossible to make a realistic practical simulation.
  • We added further work paragraph in Conclusion secsion
  • We added references to

3GPP TR 38.901 version 17.0.0,

5G NR: The Next Generation Wireless Access Technology,

I/Q Imbalance Aware Nonlinear Wireless-Powered Relaying of B5G Networks: Security and Reliability Analysis

and cited them in our work